# Phage Cocktails Constrain the Growth of *Enterococcus*

Stephen Wandro,[a] Pooja Ghatbale,[b] Hedieh Attai,[b] Clark Hendrickson,[a] Cyril Samillano,[a] Joy Suh,[a] Sage J. B. Dunham,[a] David T. Pride,[b,c] Katrine Whiteson[a]

aDepartment of Molecular Biology and Biochemistry, University of California, Irvine, California, USA
bDepartment of Pathology, University of California, San Diego, California, USA
cDepartment of Medicine, University of California, San Diego, California, USA

Stephen Wandro and Pooja Ghatbale contributed equally to this article.

**ABSTRACT** Phages that infect pathogenic bacteria present a valuable resource for treating antibiotic-resistant infections. We isolated and developed a collection of 19 *Enterococcus* phages, including myoviruses, siphoviruses, and a podovirus, that can infect both *Enterococcus faecalis* and *Enterococcus faecium*. Several of the *Myoviridae* phages that we found in southern California wastewater were from the *Brockvirinae* subfamily (formerly *Spounavirinae*) and had a broad host range across both *E. faecium* and *E. faecalis*. By searching the NCBI Sequence Read Archive, we showed that these phages are prevalent globally in human and animal microbiomes. *Enterococcus* is a regular member of healthy human gut microbial communities; however, it is also an opportunistic pathogen responsible for an increasing number of antibiotic-resistant infections. We tested the ability of each phage to clear *Enterococcus* host cultures and delay the emergence of phage-resistant *Enterococcus*. We found that some phages were ineffective at clearing *Enterococcus* cultures individually but were effective when combined into cocktails. Quantitative PCR was used to track phage abundance in cocultures and revealed dynamics ranging from one dominant phage to an even distribution of phage growth. Genomic characterization showed that mutations in *Enterococcus* exopolysaccharide synthesis genes were consistently found in the presence of phage infection. This work will help to inform cocktail design for *Enterococcus*, which is an important target for phage therapy applications.

**IMPORTANCE** Due to the rise in antibiotic resistance, *Enterococcus* infections are a major health crisis that requires the development of alternative therapies. Phage therapy offers an alternative to antibiotics and has shown promise in both *in vitro* and early clinical studies. Here, we established a collection of 19 *Enterococcus* phages and tested whether combining phages into cocktails could delay growth and the emergence of resistant mutants in comparison with individual phages. We showed that cocktails of two or three phages often prevented the growth of phage-resistant mutants, and we identified which phages were replicating the most in each cocktail. When resistant mutants emerged to single phages, they showed consistent accumulation of mutations in exopolysaccharide synthesis genes. These data serve to demonstrate that a cocktail approach can inform efforts to improve efficacy against *Enterococcus* isolates and reduce the emergence of resistance.

Address correspondence to Katrine Whiteson, katrine@uci.edu.

The authors declare no conflict of interest.

*E*nterococcus species are regular members of vertebrate gut microbiomes and are pathogens of enormous clinical significance due to their high potential for antibiotic resistance (e.g., widespread vancomycin-resistant *Enterococcus*) (1). Antibiotic treatment often leads to a high abundance of *Enterococcus* spp. in the gut, as many other members of the gut microbiome are more sensitive to antibiotics (2–5). There are multiple species of *Enterococcus* that cause human disease, but the most common are *E. faecium* and *E. faecalis*. Both *E. faecium* and *E. faecalis* develop resistance to

vancomycin, and they are known to cause disease, particularly in individuals who have undergone multiple rounds of antibiotics. The development of alternative therapies for *Enterococcus* infections is vital for providing effective treatment options for recalcitrant long-term infections, such as endocarditis.

Surprisingly, there are relatively few characterized *Enterococcus* phages. Bacteriophage (phage) therapy is an alternative treatment to antibiotics that has shown promise for treating *Enterococcus in vitro* and in animal models (6–8). However, there is a paucity of basic research into phage safety, mechanisms of action, and best practices of use. Because of the propensity for *Enterococcus* to develop antibiotic resistance, and the elevated abundance in critically ill or antibiotic-treated patients, *Enterococcus* spp. are a logical focus for phage therapy.

Although phages hold much promise for overcoming antibiotic resistance, bacteria can also evolve resistance to phage infection. Bacteria exist in a constant evolutionary battle with phages and thus have evolved many systems to resist phage infection, including preventing phage binding, restriction-modification systems, CRISPR-Cas9 immunity, and abortive infection (9, 10). Given strong selective pressure from a single phage, bacteria often quickly evolve resistance to that phage in laboratory settings (11). Despite the potential for evolved host immunity, unlike antibiotics, phages can coevolve to combat and circumvent bacterial resistance mechanisms (12). Changes that arise during coevolutionary exchanges can be tracked through experimental evolution, sequencing, and analysis of the consistently arising mutations.

Cocktails of phages are often more effective at treating infection than any one phage, because many phages have narrow host ranges. Multiple phages can be used in combination to increase the likelihood that several or all strains of the target bacteria will be killed (13, 14). Theoretically, phage cocktails also decrease the chance that a phage-resistant mutant will arise, as such a development would likely require the simultaneous evolution of multiple orthogonal resistance mechanisms. The genetic changes that lead to phage resistance can also provide a fitness disadvantage, for example, by making the bacteria more susceptible to infection by other phages (15). Similar to phage cocktails, combinations of antibiotics are used to treat tuberculosis infections, and combinations of antivirals are used to treat HIV (16, 17). The design principles for effective phage cocktails are an exciting frontier which may benefit from knowledge derived from experimental coevolution.

Phage therapy has shown promise in *in vitro* and *in vivo* mouse experiments (7, 8, 18). Additionally, *Enterococcus* phages have been used to disrupt *Enterococcus* biofilms, which are generally much harder to treat than planktonic cells, because antibiotics have trouble penetrating biofilms (8). *Enterococcus* phages have also been used to treat humans. Two phage cocktails sold by the Eliava Institute of Bacteriophages, Microbiology, and Virology in Georgia contain phages against *Enterococcus* spp. as well as phages against other common pathogens (19). *In vitro*, *Enterococcus* phage combinations have been shown to be more effective than single phages at preventing the growth of both antibiotic- and phage-resistant *Enterococcus* mutants (20). For example, Morisette et al. recently reported that, when used in combination with daptomycin, a two-phage cocktail exhibited a substantially improved capacity to eradicate *E. faecium* and prevent the emergence of phage resistance, while resistance did emerge with either phage by itself (21).While these examples are encouraging, they represent only limited examples of cocktail design. Thus, part of our goal in this work was to further compare the impact of different phage combinations on bacterial growth.

Our goals were the following: (i) characterize and test a diverse panel of phages for their ability to reduce *Enterococcus* growth; (ii) determine if and how host growth changes in the context of different phage combinations; and (iii) elucidate the underpinnings of phage resistance by identifying mutations in the *Enterococcus* host genome that are repeatedly found in the presence of phage infection.

## RESULTS

**Phage characterization and host range evaluation.** We isolated a collection of 18 *Enterococcus*-infecting phages from Southern California wastewater influent samples.

| *Enterococcus* strains | Myoviridae | | | | | | | | Siphoviridae | | | | | | | | | Podo |
| | Ben | Bop | Bob | Bill | Car | Carl | V12 | CCS1 | CCS2 | CCS3 | CCS4 | SDS1 | SDS2 | PG2 | PG9 | PG11 | PG13 | Ump |
|---|---|---|---|---|---|---|---|---|---|---|---|---|---|---|---|---|---|---|
| *E. faecium* EF18PII | ▓ | ▓ | | ▓ | | | ▓ | | | | | | | | ▓ | | | |
| EF34PII | | | | | | | | | | | | | | | | | | |
| EF41PII | ▓ | ▓ | | | | | | | | | | | | | ▓ | | ▓ | |
| EF48PII | ▓ | ▓ | | | | | ▓ | | | | | | | | | | | ▓ |
| EF50PII | ▓ | | | | | | | | | | | | | | | | | |
| EF79PII | ▓ | | | | | | | | ▓ | | | | | | | | | |
| EF98PII | ▓ | | | ▓ | | | | | | | | | | ▓ | ▓ | ▓ | ▓ | |
| TX1330 | ▓ | | | ▓ | | | | | | | ▓ | | ▓ | | | | | |
| *E. faecalis* B3286 | ▓ | ▓ | ▓ | | | | | | | | | | | | ▓ | | ▓ | ▓ |
| DP11 | ▓ | ▓ | ▓ | | ▓ | ▓ | ▓ | | | | | | | | ▓ | ▓ | ▓ | ▓ |
| DP6 | ▓ | ▓ | ▓ | | ▓ | | ▓ | ▓ | | ▓ | | | | | ▓ | ▓ | ▓ | ▓ |
| EF06 | ▓ | ▓ | ▓ | | ▓ | | ▓ | | | | | | | | | ▓ | ▓ | ▓ |
| EF07 | ▓ | ▓ | ▓ | | ▓ | | ▓ | | | | | | | | | ▓ | ▓ | ▓ |
| EF09PII | ▓ | ▓ | | | ▓ | | ▓ | | | | | | ▓ | | ▓ | ▓ | ▓ | ▓ |
| EF11 | ▓ | ▓ | ▓ | | ▓ | | ▓ | ▓ | | | | | | | ▓ | ▓ | ▓ | ▓ |
| EF116PII | ▓ | ▓ | ▓ | | ▓ | | ▓ | | | ▓ | | | | | ▓ | ▓ | ▓ | ▓ |
| Ent6 | ▓ | ▓ | ▓ | | ▓ | | ▓ | | | ▓ | | | ▓ | | ▓ | ▓ | ▓ | ▓ |
| V587 | ▓ | ▓ | ▓ | | ▓ | | ▓ | | | | | | | | ▓ | ▓ | ▓ | ▓ |
| Yi6-1 | ▓ | ▓ | ▓ | | ▓ | | | ▓ | ▓ | ▓ | ▓ | ▓ | ▓ | ▓ | ▓ | ▓ | ▓ | ▓ |

Legend: ▓ = Lysis; ☐ = No Lysis; Vancomycin-Resistant (dark gray boxes around strain names)

**FIG 1** Host ranges of *Enterococcus* phages as determined using spot assays (see Materials and Methods). Complete lysis is indicated by the orange boxes, and white boxes represent no lysis. Several vancomycin-sensitive (light gray boxes around strain names) and vancomycin-resistant (dark gray boxes around strain names) *Enterococcus* isolates were used for this study. GenBank accession numbers for the 18 additional phages discovered as part of this study along with the *Myoviridae* phage, EfV12-phi1 (V12) obtained from the Félix d'Hérelle Reference Center for Bacterial Viruses at the Université Laval are listed in Table S3 in the supplemental material.

Genome sequencing showed 8 *Myoviridae*, 10 *Siphoviridae*, and 1 *Podoviridae* phages. A list of the 18 newly isolated phages along with one already known *Myoviridae* phage, EfV12-phi1 (V12), is shown in Table S3 in the supplemental material, together with genome size and GenBank accession numbers. Plaque assays were used to test the phage susceptibilities of clinical isolates of *E. faecium* and *E. faecalis*, including both vancomycin-resistant *Enterococcus* (VRE) and vancomycin-susceptible *Enterococcus* (VSE). Eleven of 19 phages were able to lyse several strains of both *E. faecalis* and *E. faecium* (Fig. 1). Most phages from the *Myoviridae* family showed broad host ranges, especially Ben, Bop, and V12, which lysed almost all isolates.

Of the 10 *Siphoviridae* phages, 9 belonged to the genus *Saphexavirus*. Genome alignments showed that these phages share a core genome of 54 genes and approximately 43 accessory genes per phage (Fig. 2A and B). Their genomes were approximately 57 kb in length, and only 3 genes could be annotated (Fig. 2C).

Of the 8 *Myoviridae* phages, all belonged to the *Brockvirinae* subfamily, but these were divided into two genera, *Kochikohdavirus* and *Shiekvirus* (Fig. 2D). Their genomes were approximately 150 kb in length with a core genome containing ~66 genes shared among all phages in the subfamily and an accessory genome of approximately 133 genes per phage (Fig. 2E and F). The genomes were organized into two portions, shown by representative phage V12 in Fig. 2F The first portion contained shorter genes that were not always present in members of the subfamily, and the second portion contained longer genes that were more conserved among members of the subfamily. The second portion also contained the known structural genes, although most genes remained unannotated.

A major limitation for phage therapy is that phages against any specific pathogen are often not readily available. To gain a better understanding of how the phages we isolated are distributed in nature, and how common they are to isolate when hunting for candidate phages, we assessed the distribution of representative phages from the *Brockvirinae* subfamily (*Myoviridae* V12) and the *Saphexavirus* genus (*Siphoviridae* CCS3). We queried these genomes against 67,429 publicly available metagenomes in NCBI's Sequence Read Archive (Table 1) (22). Metagenomes with positive hits were downloaded and aligned to representative *Brockvirinae* genomes to ensure most of each genome was covered. *Brockvirinae* phages were found to be globally distributed in human fecal metagenomes, including those sampled in the United States, Europe, the Middle East, and Asia. *Saphexavirus* phages were only observed in two

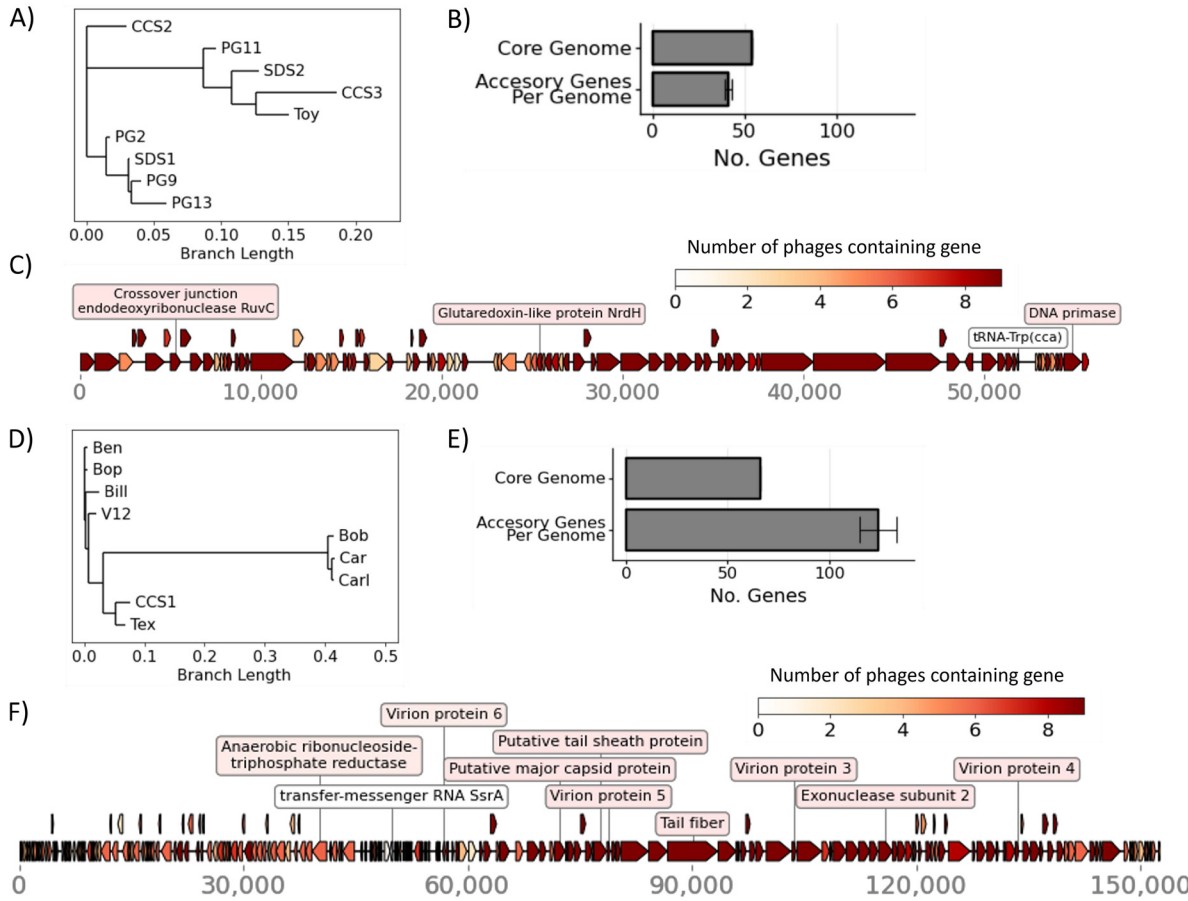

**FIG 2** Comparative genomics of major families *Myoviridae* (*Kochiodavirus* and *Schiekvirus* genera) and *Siphoviridae* (*Saphexavirus* genus). (A) Phylogenetic tree made from the core genome of *Siphoviridae* phages. (B) Number of genes in the *Siphoviridae* core genome and average number ± standard error of the mean (SEM) of accessory genes per phage genome. (C) Layout of genes in a representative *Siphoviridae* genome (phage SDS1), with known genes annotated. Colors indicate numbers of other phages that also contained each gene (50% BLASTp similarity). (D) Phylogenetic tree made from the core genome of *Myoviridae* phages. (E) Number of genes in the *Myoviridae* core genome and accessory genome per phage. (F) Layout of genes in a representative *Myoviridae* genome (phage phiV12) with known genes annotated. Colors indicate numbers of other phages that also contain each gene (50% BLASTp similarity).

metagenomes. Sequences aligning to *Brockvirinae* phages were also found in nonhuman fecal metagenomes from condors, pigs, and bats. *Brockvirinae* phages were also found to be highly abundant in two phage cocktails from the Eliava Institute designed to treat intestinal issues, the Intestiphage cocktail and the PYO phage cocktail. These phage cocktails contain many different phages targeting a wide range of bacterial hosts, including *Enterococcus*.

**Phage cocktails prevent *Enterococcus* growth longer than single phages.** Many bacterial hosts quickly develop resistance to infecting phages. Experimentally, this can be detected by resumption of bacterial growth after a period of little or no growth following phage infection. Thus, we measured bacterial growth over 72 h in liquid medium to observe whether phage cocktails could reduce bacterial growth for longer than single phages, which would suggest that cocktails are more likely to prevent resistance evolution. Four *E. faecalis* strains were chosen for testing the efficacy of phage cocktails, with strain Yi6-1 allowing for the most combinations because it was susceptible to 17 of the 18 phages in our collection (Fig. 3; see also Fig. S1 in the supplemental material). The multiplicities of infection (MOIs) of phage cocktails tested on *E. faecalis* strains, including 0.1, 0.01, and 0.001, did not consistently affect the resumption of bacterial growth (see Fig. S2). Therefore, we chose to infect strains at the highest MOI of 0.1. To ensure that the effect on host growth was not simply due to adding more

**TABLE 1** *Enterococcus* phages from the Sequence Read Archive

| Phage | SRA ID | Title | Location | Sample type |
|---|---|---|---|---|
| phiV12 | SRP077952 | INTESTI bacteriophage cocktail genome sequencing and assembly | Georgia | Phage cocktail |
| phiV12 | PRJEB23244 | PYO phage cocktail | Georgia | Phage cocktail |
| phiV12 | ERP017091 | Gut microbiome in Crohn's disease and modulation by exclusive enteral nutrition | Guangdong, China | Human fecal |
| phiV12 | ERP006678 | Gut and oral microbiome dysbiosis in rheumatoid arthritis | Beijing, China | Human fecal |
| phiV12 | SRP071229 | *Gymnogyps californianus* microbiome raw sequence reads | USA (Los Alamos National Laboratory) | California condor fecal |
| phiV12 | ERP006046 | Virus_Discovery_for_Vietnam_Initiative_on_Zoonotic_Infections__VIZIONS_ | Vietnam | Viral metagenome |
| phiV12 | ERP001956 | Diagnostic metagenomics: a culture-independent approach to the investigation of bacterial infections | Germany | Human fecal |
| phiV12 | SRP051511 | New York City MTA subway samples metagenome | USA (New York City) | Subway samples |
| phiV12 | ERP012929 | Towards personalized nutrition by prediction of glycemic responses | Israel | Human fecal |
| phiV12 | SRP040146 | *Clostridium difficile* FMT | USA (Massachusetts) | Human fecal |
| phiV12 | SRP115494 | Longitudinal multi'omics of the human microbiome in inflammatory bowel disease | USA (Massachusetts) | Human fecal |
| phiV12 | SRP099123 | Metagenomic analysis of gut microbiota in sows and piglets | Freie University of Berlin | Pig fecal |
| CCS3 | SRR1438030 | Metagenomic identification of novel enteric viruses in urban wild rats and genome characterization of a group A rotavirus | Berlin, Germany | Rat fecal |
| CCS3 | ERR2737461 | Virus discovery for Vietnam Initiative on Zoonotic Infections (VIZIONS) | Vietnam | Viral metagenome |

phage, the amount of each phage was halved relative to the single-phage condition when two phages were used and reduced to one-third when three phages were used.

Individual phages and phage cocktails had varying success in preventing host growth over 72 h, with some phages and cocktails preventing growth and others failing to do so (Fig. 3). Successful two-phage cocktails were usually composed of multiple phage families (for instance, *Myoviridae* and *Siphoviridae*), but not all cocktails composed of two different families were successful at preventing growth. Several successful two- and three-phage cocktails against *E. faecalis* Yi6 included phages that were unable to prevent host growth alone, suggesting that synergy between the phages was responsible for limiting host growth. All combinations of two- and three-phage cocktails against *E. faecalis* strain V587 (VRE) were successful in preventing host growth. *E. faecalis* strain V587 was only susceptible to *Myoviridae* phages in our cocktails showing that synergy can occur within the same family of phages.

**Phage growth dynamics by qPCR.** The dynamics of phage growth and phage-phage interactions in a cocktail were examined by quantitative PCR (qPCR), which provides an estimate of the relative abundance of each phage throughout infection. We formulated cocktails containing two or three phages that were ineffective alone but successful together in preventing bacterial growth over 72 h, and we chose *E. faecalis* strain Yi6-1 as the host. Phage DNA was extracted immediately after phage and host inoculation (0 h), as well as at 24, 48, and 72 h postinoculation. Changes in phage abundances over time in each of eight cocktails were compared with a phage-only control that contained no bacteria. *E. faecalis* strain Yi6-1 growth without any phage is shown for comparison, as well as results for each individual phage infecting *E. faecalis* strain Yi6-1 alone (Fig. 4A and B).

When grown in culture without any phage, *E. faecalis* strain Yi6-1 grew to an optical density at 600 nm ($OD_{600}$) of 0.7. When these individual phages were added (remembering that these phages were chosen because they failed to totally inhibit growth), growth was inhibited to an $OD_{600}$ of ~0.15 on average (Fig. 4B). When used in two- or three-phage cocktails, the $OD_{600}$ was reduced to nearly zero (Fig. 4A and C to J). Using qPCR to measure phage abundances, we determined which phages were contributing most to bacterial growth inhibition.

We monitored the concentration of each phage in two- and three-phage cocktails against *E. faecalis* strain Yi6-1 (Fig. 4C to J). In seven of eight of the cocktails, the

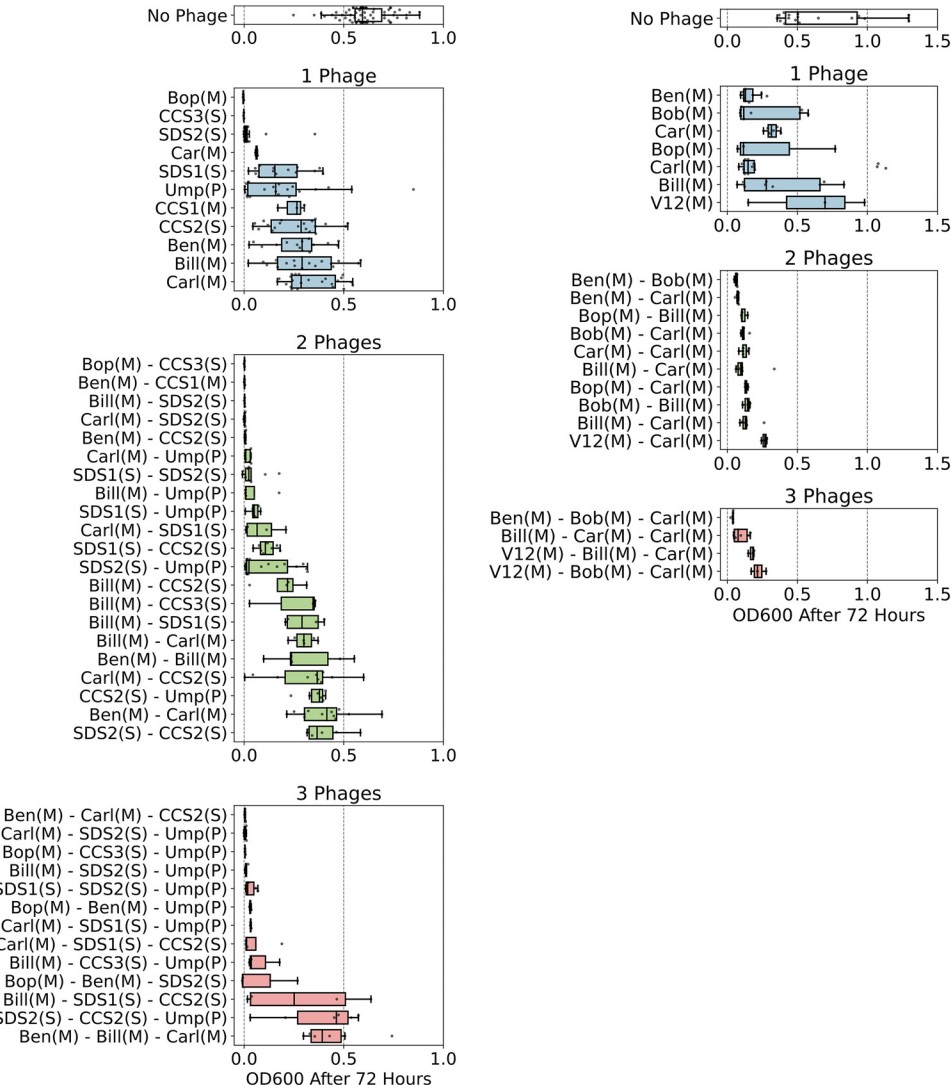

**FIG 3** Phage cocktails cleared cultures and prevented growth of *E. faecalis* for 72 h. Data points indicate the final $OD_{600}$ of replicate bacterial cultures after 72 h of incubation with phage. Boxplots represent medians and interquartile ranges. Combinations of one, two, or three phages were added to susceptible *E. faecalis* Yi6-1 (A) and *E. faecalis* V587 (B) cultures in exponential growth phase, and mixtures were incubated for 72 h. M, *Myoviridae* phage; S, *Siphoviridae* phage; P, *Podoviridae* phage.

abundances of the phages were similar, suggesting that all phages present were contributing to bacterial lysis at a similar level. The one exception was in two-phage cocktail 6, in which phage Ump showed nearly a 6-fold increase in abundance over phage Carl (Fig. 4H).

**Phage infection selects for mutations in *Enterococcus* exopolysaccharide synthesis genes.** Host mutations that emerge across replicates after phage infection indicate genes with the potential to confer phage resistance. To identify these genes, we sequenced several strains of *Enterococcus* that could grow in the presence of phage infection. These included *E. faecalis* strains DP11, Yi6, B3286, and TX2137 as well as *E. faecium* strain TX1330. Target strains were also grown in identical conditions in the absence of phage to ensure that observed mutations were specific to evolution with phages. Sequencing reads were compared to the original strain genome to determine if any genes had acquired mutations. All observed mutations are summarized in Table 2, with the position of mutations within the Epa locus shown in Fig. S3 in the

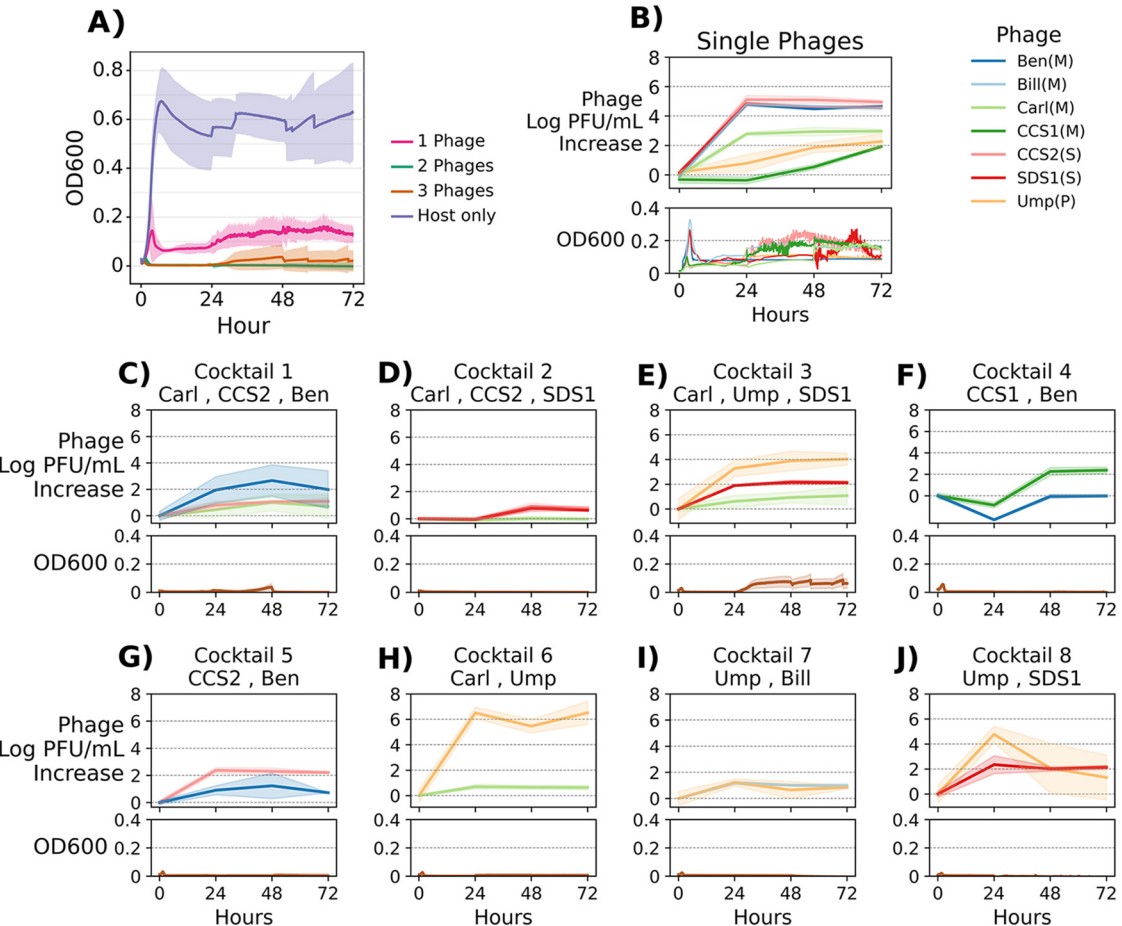

**FIG 4** Phage abundances and *E. faecalis* Yi6-1 growth with phage cocktails. Phage was added at hour 0, and the cultures were grown for 72 h. Individual phage abundances were measured by qPCR and plotted as the replicate mean ± SEM of the log PFU per milliliter increase from hour 0. (A) Bacterial growth when *E. faecalis* Yi6-1 was grown alone (host only), with each phage separately (1 phage), or with phage cocktails (2 phages, 3 phages). The replicate mean ± SEM $OD_{600}$ is shown. (B) Each phage was added separately to growing *E. faecalis* Yi6-1 cultures. The phage abundance is plotted in the top panel, and the bacterial growth curve is shown in the bottom panel. (C to J) Phage cocktails were added to growing *E. faecalis* Yi6-1 cultures. The abundance of each phage cocktail is plotted in the top panel, and the bacterial growth curve is plotted in the bottom panel.

supplemental material. *Enterococcus* strains consistently acquired point mutations in genes involved in exopolysaccharide synthesis. The four *E. faecalis* strains acquired mutations in the Epa exopolysaccharide synthesis locus, while *E. faecium* TX1330 acquired mutations in the Yqw exopolysaccharide synthesis locus. In the host-only controls, mutations in the Epa locus were not observed; however, we did observe Yqw mutants, indicating that these mutations may arise without the addition of lytic phage.

## DISCUSSION

We set out to discover phages that can infect *Enterococcus* spp., characterize their host ranges and genomes, evaluate their interactions as single phages and in cocktails, and use experimental coevolution to define where selection pressures lead to accumulation of functionally equivalent mutations. Several of the myoviruses we found in Southern California wastewater were from the *Brockvirinae* subfamily (formerly *Spounavirinae*), and were able to infect both *E. faecium* and *E. faecalis*. To better understand the biology of these phages in the context of phage therapy, we examined their prevalence in public sequence data. We evaluated fecal metagenomes and found that *Brockvirinae* phages are globally distributed in animal and human microbiomes. We also showed that phage infection dynamics change in the presence of multiple phages and that cocktails of phages restrict bacterial growth for at least 72 h.

**TABLE 2** Mutations observed in *Enterococcus* genomes after culturing with individual phages[a]

| Enterococcus strain | Phage(s) | Mutated host gene | Locus ID | Mutation type | Exopolysaccharide locus | V587 locus tag (EPA only) | AA change |
|---|---|---|---|---|---|---|---|
| *E. faecalis* DP11 | Ump, SDS1 | NAD-dependent epimerase/dehydratase | HOCGOLEH_00595 | SNP | Epa gene | EF_2165 | G11V, G279E |
| *E. faecalis* DP11 | Ump | Bacterial sugar transferase | HOCGOLEH_00585 | Nonsense | EpaR | EF_2177 | E249* |
| *E. faecalis* DP11 | SDS2 | TagF gene. glycerol glycerophosphotransferase | HOCGOLEH_00593 | SNP | Epa gene | | A113E |
| *E. faecalis* Yi6 | Bop | Bacterial sugar transferase | UMS_01916 | SNP, nonsense | EpaR | EF_2177 | N705Y |
| *E. faecalis* Yi6 | Bop | UTP-glucose-1-phosphate uridylyltransferase | UMS_01646 | DEL | | | K22* |
| *E. faecalis* Yi6 | Bop | UDP-*N*-acetylglucosamine 1-carboxyvinyltransferase 1 | UMS_01161 | SNP | | | A210T |
| *E. faecalis* Yi6 | Bop | Isoleucyl-tRNA synthetase | UMS_00984 | DEL | | | W31* |
| *E. faecalis* Yi6 | phiV12 | 30S ribosomal protein S7 | UMS_00243 | SNP | | | G82D |
| *E. faecalis* Yi6 | phiV12 | DNA-directed RNA polymerase subunit alpha | UMS_00275 | SNP | | | G29V |
| *E. faecalis* B3286 | Bop, phiV12 | NAD-dependent epimerase/dehydratase | SQ1_02166 | SNP, nonsense | Epa gene | EF_2165 | A123E, Q204* |
| *E. faecalis* B3286 | Bop | Glucose-1-phosphate thymidylyltransferase; | SQ1_02191 | SNP | EpaE | EF_2194 | R217C |
| *E. faecalis* B3286 | Bop | Endonuclease III | SQ1_01204 | SNP | | | A130V |
| *E. faecalis* B3286 | phiV12 | Bacterial sugar transferase | SQ1_02177 | SNP | EpaR | EF_2177 | Y336* |
| *E. faecalis* B3286 | phiV12 | ATP-dependent Clp protease ATP-binding subunit ClpE | SQ1_00765 | SNP | | | I13F |
| *E. faecalis* TX2137 | Bop | Epimerase/dehydratase | HMPREF9494_02361 | SNP | EpaW | EF_2171 | D229Y, G192V |
| *E. faecalis* TX2137 | Bop | Phosphocarrier protein HPr | HMPREF9494_02513 | SNP | | | V209L |
| *E. faecalis* TX2137 | Bop, phiV12 | Bacterial sugar transferase | HMPREF9494_02367 | SNP | EpaR | EF_2177 | R400C, G251R |
| *E. faecalis* TX2137 | phiV12 | DNA ligase (NAD$^+$) | HMPREF9494_02501 | SNP | | | A434E |
| *E. faecium* TX1330 | Ben | DNA gyrase subunit A (EC 5.99.1.3) | HMPREF0352_0587 | SNP | | | D116A |
| *E. faecium* TX1330 | Ben | DNA-directed RNA polymerase beta' subunit (EC 2.7.7.6) | HMPREF0352_2730 | SNP | | | A924V |
| *E. faecium* TX1330 | Bob | Response regulator transcription factor | D3Y30_RS07660 | SNP | | | E192K |
| *E. faecium* TX1330 | Bop | Polysaccharide biosynthesis protein | D3Y30_RS11110 | SNP | | | L222S |
| *E. faecium* TX1330 | Bop, Ben, Bill, no phage | CpsD/CapB family tyrosine-protein kinase | D3Y30_RS11120 | SNP | Yqw | | P136S, P26Q, L62E |
| *E. faecium* TX1330 | Carl, phiV12, no phage | Tyrosine protein kinase | D3Y30_RS11125 | SNP | Yqw | | A147T, V153A |

[a]Each row indicates an *Enterococcus* gene in which one or more mutations were observed when that *Enterococcus* strain was cultured with the indicated phage. When more than one phage is listed the mutation occurred separately for each phage host pair and the phages were not used in combinations. Locus ids refer to the genomes listed in Table S2 of the supplemental material. SNP, single-nucleotide polymorphism; DEL, deletion.

While using phage cocktails that include a diversity of phages targeting multiple disparate bacterial types has been a common practice in phage therapy, there is no standard for how many phages should be used, and the answer will likely change for different bacterial hosts. Some phage cocktails are effective at killing bacteria and preventing growth of resistant mutants, but the most effective phage combination is unknown for the vast majority of pathogenic bacteria. For example, the pyophage (PYO) cocktail from the Georgian Eliava Institute of Bacteriophages contains approximately 30 different phages targeting multiple bacterial hosts (19). Recent uses of phage therapy designed to target a single strain of a bacterium have included between one and six phages (23–25). Often, there is no obvious rationale behind the number of phages chosen for phage therapy, although lack of access to phages against the specific host of interest is often a limitation. Here, we show that combinations of two phages are effective at inhibiting growth of several *Enterococcus* strains over 72 h. Theoretically, using more phages in a cocktail would increase the chances of choosing phages that displayed synergy in reducing the host growth. However, increasing the number of phages could also lead to antagonistic interactions between phages (26). Antagonistic interactions may arise from competition for finite host resources, competition for host receptors, or the production of phage repressors.

Some phages in our collection were effective against *E. faecalis* and *E. faecium*, including both VRE and VSE isolates, suggesting that they would be good candidates for phage cocktails against a broad range of *Enterococcus* spp. If the phages in a cocktail target different bacterial proteins as binding sites, the cocktail will be more effective at lysing a pathogen even if a mutation in a single protein arises (27). As shown in Fig. 3A, combinations of phages that individually cannot clear a culture of *E. faecalis* Yi6-1 are able to do so when used in combinations. This shows that the evolutionary advantage of phage-resistant mutants could be diminished with the use of well-designed cocktails (28, 29).

Comparing the abundances of each phage over time in the eight phage cocktails shown in Fig. 4 yields a wide range of outcomes, from one phage greatly outpacing the other (cocktail 6), to relatively even abundances (cocktails 1, 7, and 8). Each of the five phages was ineffective at clearing *Enterococcus* cultures alone, yet all cocktails, with their varied phage dynamics, resulted in clearing of the *Enterococcus* cultures. This shows that there is more than one path to an effective phage cocktail. Even if one phage appears to dominate, the other phage may still be necessary to prevent the emergence of a phage-resistant mutant.

When under selective pressure from *Brockvirinae* phages, *Enterococcus* produce mutations primarily in exopolysaccharide synthesis genes, suggesting that phage resistance may evolve by preventing phage recognition and initial binding. *E. faecium* and *E. faecalis* both contain the highly conserved Epa capsule synthesis locus, in which genetic variation has been observed consistently for *E. faecalis* strains (30, 31, 32). Mutations in the Epa locus have been observed previously during coevolution with *Brockvirinae* phages; these mutations impaired *Enterococcus* host colonization and increased antibiotic sensitivity (7, 33). Consistent with the results presented herein, Canfield et al. observed mutations to the *E. faecium epaR* and *epaX* genes during bacteriophage infection (34, 35). In addition to the Epa locus, *E. faecium* encodes the Yqw exopolysaccharide synthesis locus, which is not present in *E. faecalis*. Mutations were observed in the Yqw locus of *E. faecium* TX1330, but not in the Epa locus. Reproducible mutations arising across replicates within genes of the Yqw locus were previously observed during coevolution between phage and *Enterococcus* (36). However, in the current experiments, mutations in Yqw locus genes occurred in both TX1330 host control cultures (which lacked phage) and cultures containing phage. Therefore, we cannot attribute these mutations to phage evolutionary pressure.

The need for alternative therapies for antibiotic-resistant bacterial infections continues to grow. Bacteriophages present an alternative therapeutic route with the potential to replace or supplement antibiotics, but there is a significant knowledge gap that

reduces the utility of phages. Challenges facing phage therapy include the emergence of host resistance, limited information on how phages may collectively eradicate their hosts, and a lack of guidance on the number of phages necessary for the formulation of successful phage cocktails. *Enterococcus* is a good model bacterium because it is a diverse genus containing multiple species capable of causing debilitating human infections, and it has the ability to acquire significant antibiotic resistance. Using this model, we have demonstrated that simple two-phage cocktails have significant potential to kill their hosts and reduce the emergence of resistant isolates. For *Enterococcus*, our data suggest that host killing was not substantially increased by adding more than two phages to cocktails, but a larger study is needed to confirm these findings. Rationally designing phage cocktails with knowledge of phage-host interactions from experimental evolution has the potential to significantly advance phages as antibiotic alternatives for the treatment of human pathogens such as *Enterococcus*.

## MATERIALS AND METHODS

**Bacterial strains and culture conditions.** The bacterial strains used in this study are listed in Table S2 of the supplemental material. *E. faecalis* and *E. faecium* strains were added to our strain collection from the patient population at UC San Diego Health; all personally identifying information was removed. Each of these isolates was identified using a biotyper instrument based on matrix-assisted laser desorption ionization–time of flight spectroscopy technology (Bruker, Billerica, MA, USA), and antibiotic susceptibilities were assessed using broth microdilution techniques on the BD Phoenix instrument (Becton, Dickinson, Franklin Lakes, NJ, USA). Four of the strains were obtained from the Human Microbiome Project repository at the Biodefense and Emerging Infections Research Resources Repository (BEI: www.beiresources.org). *E. faecalis* and *E. faecium* strains were cultured in brain heart infusion (BHI) broth. All strains were grown at 37°C in liquid medium overnight with shaking. Solid medium was prepared with 1.5% agar when culturing bacteria or 1.0% bottom agar and 0.3% top agar for plaque assays.

**Phage isolation propagation and storage.** Phages were isolated from sewage using three rounds of plaque assays. Raw sewage influent was collected from wastewater treatment plants in Orange County, Redwood Shores, and Escondido, California. Sewage was stored at 4°C and used for phage isolation for several months. Sewage was centrifuged for 10 min at 10,000 × *g* to remove particulates, and the supernatant was used in plaque assays with various strains of *Enterococcus*. A 100-μL aliquot of sewage supernatant was added to 100 μL exponentially growing *Enterococcus* in BHI medium and incubated at 37°C for 15 min. Five milliliters of warm BHI containing 0.3% UltraPure low-melting-point agarose (ThermoFisher catalog number 16520050) was added, and the mixture was poured on a BHI agar plate for overnight incubation at 37°C. The next day, plates were examined for plaques, and any plaques were picked with a pipette tip and suspended in 50 μL SM buffer (37). Picked plaques underwent two more rounds of plaque assays in the same manner to ensure purity of the phage isolate. Pure phages were propagated by performing a plaque assay to create a plate displaying webbed lysis that was then flooded with 3 mL of SM buffer and incubated for 1 h. The SM buffer was then collected and centrifuged at 10,000 × *g* for 10 min. For long-term storage, phages were stored at −80°C in SM buffer containing 25% glycerol.

**Genomic sequencing.** DNA was extracted from *Enterococcus* and phage using a Quick-DNA Microprep kit (Zymo catalog number D3020). Before *Enterococcus* DNA extraction, lysozyme was added to lysis buffer at a concentration of 100 μg/mL and incubated at 37°C for 30 min. For DNA extraction from coevolution cultures containing both bacteria and phage, the extractions were performed without lysozyme. Libraries were prepared using scaled-down reactions with the Illumina Nextera enzyme (24). Paired-end sequencing with a 75-bp read length was performed on the Illumina NextSeq using the Mid Output v2 reagents. Approximately 2.5 million reads were obtained for each sample.

**Genomic characterization.** Phage and *Enterococcus* genomes were assembled *de novo* using the SPAdes assembler (38). Genomes were annotated using Prokka with the genus set to *Caudovirales* (39). Core genomes were determined and aligned using Roary (40). Phylogenetic trees were constructed from core genomes using FastTree (41). Visualizations were made using the Python matplotlib, dna_features_viewer, and Biopython (42).

**Searching the Sequence Read Archive.** All metagenomes in the SRA were searched for *Brockvirinae Enterococcus* phages using the "searching SRA" tool with V12 and CCS3 as representative phages for the myovirus and siphovirus families, respectively (22). Briefly, the searching SRA tool searches for the query sequence in all 111,156 metagenomes currently on the SRA by subsampling 100,000 sequences from each metagenome. From the metagenome hit list, we selected only metagenomes where the average read length matching our query was over 50 bp. Alignments were manually inspected using Geneious to ensure the majority of the genome was covered (43). Information about the SRA projects with verified hits is summarized in Table 1.

**Host range determination.** Phage susceptibilities were measured in several bacterial strains with established multidrug resistance serious enough to cause illness (Fig. 1). Phage susceptibility was determined using a spot assay in which 5 μL of each phage lysate was spotted on a lawn of an *Enterococcus* strain on a 1.5% agar plate infused with BHI. The spots were allowed to dry at room temperature for 30 min and incubated at 37°C for 24 h. The next day, plates were examined to identify the host's susceptibility based on the size and shape of the cell lysis zones.

Although spot assays may show lysis from without due to too many phages infecting the cell simultaneously, this is not a widespread phenomenon (44). Furthermore, our MOI experiments, and also our observation of concentration-dependent host lysis with different phage titers in liquid and on plates (see Fig. S2 in the supplemental material), indicated that lysis from without was not occurring in the spot assays. Nevertheless, we cannot completely exclude the possibility that lysis from without contributed to some of the spot assay results.

**Determination of MOI.** MOI is defined as the ratio of the number of phage particles to the number of target cells of each host. We tested three MOIs (0.1, 0.01, and 0.001) as follows. A single colony was picked from a streak plate and was grown overnight in BHI broth at 37°C. The next day, the bacterial culture was diluted to an $OD_{600}$ of 0.05 in fresh BHI. A serial dilution series from $10^{-1}$ to $10^{-8}$ of each bacterial culture was performed using phosphate-buffered saline. For each isolate, an aliquot from serial dilutions of $10^{-4}$ and $10^{-5}$ was plated on 1.5% BHI agar plates and incubated overnight at 37°C. The next day, the number of colonies was counted on each dilution plate to determine the number of CFU per milliliter (see Table S1 in the supplemental material). Host susceptibility to each phage was determined via plaque assays. Cells from the log phase of growth were infected with different phages at different dilutions. Based on plaque assay plates, we determined the titer of each phage (in PFU per milliliter) (see Table S1). Based on observed CFU of *Enterococcus* isolates and the titer of each phage, the appropriate volume of isolates and phages were determined at three different MOIs, 0.1, 0.01, and 0.001, to conduct growth curve experiments.

**Phage cocktails.** Cocktails consisting of one, two, or three phages were tested against *E. faecalis* Yi6-1, EF06, EF11, and V587. A single colony was inoculated into BHI broth and grown overnight in a shaker at 37°C. The next day, bacterial cultures were diluted to an $OD_{600}$ of 0.05 in fresh BHI. Based on the results of three MOIs (0.1, 0.01, and 0.001) (see Fig. S2 in the supplemental material), appropriate volumes of *Enterococcus* and phages were determined and combined in a single well inside a 96-well plate along with enough BHI to make up a total volume of 200 $\mu$L. For two-phage cocktails, one-half of the previously determined volumes of each of the two unique phage stocks were added. To evaluate three-phage cocktails, one-third of previously determined volumes of each of the three unique phage stocks was added. To avoid desiccation in wells at the perimeter of the 96-well plates, 200 $\mu$L of fresh BHI medium was added to all perimeter wells.

**PCR.** Phage cocktails were grown in BHI for 72 h with their respective hosts (Yi6-1, EF06, and EF11) in 96-well plates with three biological replicates. The positive control consisted of phages without a host. The negative controls consisted of the relevant hosts grown in BHI for 72 h without phage. The samples were collected at four time points (0 h, 24 h, 48 h, and 72 h), and total genomic DNA was isolated using the DNeasy blood and tissue kit (Qiagen). For the unknown standards, we performed a plaque assay for all the phages to find their concentrations (in PFU per milliliter), and the genomic DNA of the standards was isolated using the DNeasy blood and tissue kit. Standard curves were generated with serial dilutions of phage ($10^{-1}$ to $10^{-8}$). Phage-specific primers were designed using Geneious software and are listed in Table S4 of the supplemental material. For designing the primers, all the phages were aligned using the multiple alignment tool in Geneious software, and unique regions within each phage were selected. The qPCR experiment was performed in a 96-well PCR plate using an Eppendorf Mastercycler RealPlex with SYBR green PCR master mix (Eppendorf) as per the universal SYBR green qPCR protocol, where fluorescent product is detected during the last step of each cycle. The obtained melting curve data were analyzed using Eppendorf Mastercycler RealPlex to calculate the Cycle threshold (Ct) values. Ct values of the standards were then used to generate standard curves correlating the log PFU per milliliter to Ct values; this information was used to estimate the concentration of each phage in the cocktails. Changes in PFU per milliliter were plotted over time for each phage cocktail.

**Generation of *Enterococcus* mutants in liquid cultures.** To determine if apparent phage resistance was associated with genetic mutations, we grew five *Enterococcus* spp. strains in the presence of different phages and sequenced those that displayed a resistance phenotype. Both *E. faecalis* (B3286, TX2137, Yi6, and DP11) and *E. faecium* (TX1330) hosts were grown individually (only when the host was susceptible by spot assay) with Ben, Bop, Bill, Bob, Carl, EfV12-phi1, SDS2, and Ump. *Enterococcus* sp. strains were grown overnight in BHI, diluted to an $OD_{600}$ of 0.05, aliquoted into 96-well plates with 10 $\mu$L of a highly concentrated individual phage stock (total volume of 100 $\mu$L), and incubated at 37°C. Every 24 h for 28 days, 10 $\mu$L of culture was diluted into 190 $\mu$L fresh BHI media. Wells that showed a resurgence of growth, potentially indicating evolved resistance, were frozen at −80°C in 50% glycerol for DNA extraction and sequencing.

**Enterococcus mutant sequencing.** DNA was extracted from *Enterococcus* cultures that appeared to be resistant to phages by using the Quick-DNA Microprep kit (Zymo catalog number D3020). Before *Enterococcus* DNA extraction, lysozyme was added to lysis buffer at a concentration of 100 $\mu$g/mL and incubated at 37°C for 30 min. Libraries were prepared using a scaled-down protocol with the Illumina Nextera enzyme (45). Short-read-length (75 bp) paired-end sequencing was performed on the Illumina NextSeq using the Mid Output v2 reagents. Approximately 1 million reads were obtained per sample, resulting in about 10-fold coverage across the *Enterococcus* genome. All sequencing experiments were performed using cultured populations of bacteria, as opposed to individual colony-purified strains; therefore, each culture likely contained DNA from multiple strain variations.

**Sequencing analysis.** SPAdes was used for genome assembly for phages and bacteria when reference genomes did not already exist (38). For calling bacterial mutations, DNA sequencing reads from each phage-resistant host population were aligned to their wild-type genome using Breseq, which performs short-read alignment to a reference and calls mutations (46). In cases where an existing assembly was used, preexisting mutations at the initial time point were subtracted. All mutations are reported in Table 2. To relate mutations in exopolysaccharide synthesis genes in the Epa locus among strains, genes

were mapped to the *E. faecalis* V583 genome (GenBank GCF_000007785.1), in which the Epa locus has been well-characterized (47).

**Data availability.** Data from bacterial growth assays, phage qPCR, and code for analysis and making figures are available at https://github.com/swandro/phage_cocktails. Genomes for bacterial and phage strains used in this study have been deposited with GenBank, and the available accession numbers for all phage and some bacteria can be found in Tables S2 and S3 in the supplemental material.

## SUPPLEMENTAL MATERIAL

Supplemental material is available online only.

**FIG S1**, TIF file, 0.3 MB.
**FIG S2**, TIF file, 0.4 MB.
**FIG S3**, TIF file, 0.2 MB.
**TABLE S1**, DOCX file, 0.4 MB.
**TABLE S2**, DOCX file, 0.02 MB.
**TABLE S3**, DOCX file, 0.02 MB.
**TABLE S4**, DOCX file, 0.01 MB.

## ACKNOWLEDGMENTS

We acknowledge the Orange County Sanitation District, along with the wastewater treatment facilities in Redwood Shores, CA, and Escondido, CA, for providing influent samples from which some of the phages were isolated; the Félix d'Hérelle Reference Center for Bacterial Viruses at the Université Laval for providing phage V12, Dr. Heather Maughan for meaningful edits, a T32 training grant to Stephen Wandro (T32AI007319); an R21 awarded to Katrine Whiteson and David Pride (R21AI149354); and the UC San Diego Health Clinical Microbiology Laboratory and Peiting Kuo for performing antibiotic susceptibilities.

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
