## [Reviewer comments · mSystems]

Phage Cocktails constrain the growth of Enterococcus

Stephen Wandro, Pooja Ghatbale, Hedieh Attai, Clark Hendrickson, Cyril Samillano, Joy Suh, Sage Dunham, David Pride, and Katrine Whiteson

Corresponding Author(s): Katrine Whiteson, University of California, Irvine

Review Timeline:

Submission Date:	January 14, 2022
Editorial Decision:	February 2, 2022
Revision Received:	April 28, 2022
Accepted:	May 9, 2022

Editor: Marta Gaglia

Reviewer(s): Disclosure of reviewer identity is with reference to reviewer comments included in decision letter(s). The following individuals involved in review of your submission have agreed to reveal their identity: Andrew Camilli (Reviewer #3)

Transaction Report:

DOI: <https://doi.org/10.1128/msystems.00019-22>

February 2, 2022

Dr. Katrine Whiteson
University of California, Irvine
Department of Molecular Biology and Biochemistry
School of Biological Sciences
3236 McGaugh Hall
Irvine, CA 92697-3900

Re: mSystems00019-22 (Phage Cocktails constrain the growth of Enterococcus)

Dear Dr. Katrine Whiteson:

Thank you for submitting your manuscript to mSystems. We have completed our review and I am pleased to inform you that, in principle, we expect to accept it for publication in mSystems. However, acceptance will not be final until you have adequately addressed the reviewer comments.

Please also note that all the sequence and genomics data needs to be deposited in appropriate databases prior for the final acceptance.

Preparing Revision Guidelines

Sincerely,

Marta Gaglia

Editor, mSystems

Journals Department
Reviewer comments:

Reviewer #1 (Comments for the Author):

The revised manuscript by Wandro et al, has nicely addressed my previous concerns and questions. I have two remaining comments for the authors.

1. The authors now show various mutations that arose in *E. faecium* following phage predation. A recent paper identified numerous polysaccharide, cell wall, and replication gene mutations associated with *E. faecium* phage resistance (Canfield et al. AAC. 2021 <https://pubmed.ncbi.nlm.nih.gov/33649110/>). The authors should discuss their discovery of *E. faecium* mutations that result in phage resistance within the context of this recently published work.
2. Additionally, combination phage treatment with and without dapto exposure using *E. faecium* has recently been studied (Morrisette et al. AAC. 2022 <https://pubmed.ncbi.nlm.nih.gov/34723631/>). The authors should reference to this work to support their narrative surrounding phage cocktails.

Reviewer #3 (Comments for the Author):

Major comments

1. Table 2 is fairly uninformative as is, and is in need of major improvement. A SNP has little meaning by itself. The SNP and position in the gene should be listed (eg., A225G, or G112-, etc.). Figure S3 is not sufficient to provide this information. Instead of SNP, use missense, nonsense, silent, DEL, or INS. If a missense mutation, then the amino acid change and residue number should be listed (eg., Ala48Gln). The table length will likely expand, since some genes may have multiple different mutations. However, this is very important data that is central to the paper.
2. At least a few of the phage resistant mutants in Table 2 should be validated by plaque assay using each of the phages. The story is incomplete without this data.

Minor comments

1. Line 90: Change "would" to "may". If the phages in the cocktail use the same receptor, then a single mutation can give resistance to all the phages.
2. Line 92: Change "making" to "for example making". It is more common that phage resistance mutations reduce virulence or some other facet of fitness than resistance to other phages. In fact, you make this point in the Discussion, Line 282.
3. Line 103 and elsewhere: Don't italicize "spp".
4. Line 105: Antibiotic resistant or phage resistant? Clarify.
5. Lines 117-124: You don't say if the 20 phages are novel or already known. If already published or their genome is in Genbank, you need to reference accordingly.
6. Line 125: Delete "Core". It is must be "Genome alignments.." since you wouldn't know what is a core gene prior to aligning the genomes.
7. Line 140: Change "how phages" to "how select phages".
8. Fig 2: Some numbers on the trees overlap the tree, making it hard to decipher. Move them off to one side. If necessary, cover them up and retype the number in a better location. Also, some font sizes are very tiny. Make larger.
9. Fig 4A: Are the lines in the graph showing the mean and SD of all of the phage treated cultures? For instance, is the line for the 1 phage cultures the mean and SD of all growth curves with 1 phage? This needs to be explained in the figure legend.
10. Line 312: Are you saying the patient's names were de-identified? Presumably not, since that would be a HIPAA violation? Perhaps you mean the clinical isolates of *Enterococcus* were de-identified.
11. Line 453: It appears you prepped and sequenced gDNA from the culture populations, instead of from individual colony-purified and confirmed phage-resistant strains. Is this true? If so, it should be stated that the cultures may contain multiple strains for clarification.

Response to reviewer comments for “Phage Cocktails constrain the growth of *Enterococcus*” by Stephen Wandro, Pooja Ghatbale, and coauthors (# mSystems00019-22R1):

We thank the reviewers for the time spent reviewing our manuscript and for their helpful suggestions. Their comments have made the manuscript much better during both revisions. We’ve made several changes to the manuscript text and figures. See below for a point-by-point response to the reviewer’s comments (in purple font) and a detailed guide to the manuscript changes.

Response to Reviewer #1:

1. The authors now show various mutations that arose in *E. faecium* following phage predation. A recent paper identified numerous polysaccharide, cell wall, and replication gene mutations associated with *E. faecium* phage resistance (Canfield et al. AAC. 2021 <https://pubmed.ncbi.nlm.nih.gov/33649110/>). The authors should discuss their discovery of *E. faecium* mutations that result in phage resistance within the context of this recently published work.

Response: Thank you for highlighting the research of Canfield et al.. We’ve now discussed the relevant findings in our manuscript. Importantly, we have revised Table 2 to include a column detailing the *Epa* Locus so that it is possible to relate our findings, both internally between the strains we included and with the findings of other papers. We both see mutations in the *EpaR* gene, in our case in 4 different *Enterococcus faecalis* strains: Yi6, DP11, B3286, and TX2137. Specifically, starting on Line 288 of the discussion, we highlight the fact that many of the phage-induced mutations that we observe arise in the *epaR* gene of the exopolysaccharide synthesis locus, matching the findings of Canfield et al.. “Consistent with the results presented herein, Canfield *et. al.*, observed mutations to the *E. faecium epaR* and *epaX* genes during bacteriophage infection.”

2. Additionally, combination phage treatment with and without dapto exposure using *E. faecium* has recently been studied (Morrisette et al. AAC. 2022 <https://pubmed.ncbi.nlm.nih.gov/34723631/>). The authors should reference to this work to support their narrative surrounding phage cocktails.

Response: Thank you for bringing this manuscript to our attention. The findings of Morrisette et al. (i.e. that phage cocktails in combination with the antibiotic daptomycin prevent the emergence of phage resistance) closely support those in our own report. We have incorporated a discussion of the relevant findings into our manuscript. Specifically, starting on page 107 of the manuscript we write: “For example, Morisette *et al.*, recently reported that, when used in combination with daptomycin, a two-phage cocktail exhibited a substantially improved capacity to eradicate *E. faecium* and prevent the emergence of phage resistance, while resistance does emerge with either phage by itself. While these examples are encouraging, they represent only limited examples of cocktail design.”

Response to Reviewer #2 (Major Comments):

1. Table 2 is fairly uninformative as is, and is in need of major improvement. A SNP has little meaning by itself. The SNP and position in the gene should be listed (eg., A225G, or G112-, etc.). Figure S3 is not sufficient to provide this information. Instead of SNP, use missense, nonsense, silent, DEL, or INS. If a missense mutation, then the amino acid change and residue number should be listed (eg., Ala48Gln). The table length will likely expand, since some genes may have multiple different mutations. However, this is very important data that is central to the paper.

Response: We agree, and have substantially modified Table 2 with the suggested additions, including a column focused on the Epa locus as described in Reviewer 1 Response 1 above.

2. At least a few of the phage resistant mutants in Table 2 should be validated by plaque assay using each of the phages. The story is incomplete without this data.

Response: Thank you for this suggestion. Although we agree that this line of experiments would be informative, colony purified isolates would be required for effective plaque assays, and obtaining these isolates is beyond the scope of our manuscript. As addressed in point 11 below, all sequencing experiments were performed on communities, not bacterial isolates.

Response to Reviewer #2 (Minor Comments):

1. Line 90: Change "would" to "may". If the phages in the cocktail use the same receptor, then a single mutation can give resistance to all the phages.

Response: Thank you for the suggestion. To address this point we changed "would" to "would likely" in Line 90.

2. Line 92: Change "making" to "for example making". It is more common that phage resistance mutations reduce virulence or some other facet of fitness than resistance to other phages. In fact, you make this point in the Discussion, Line 282.

Response: "for example by" was added to line 92 as suggested.

3. Line 103 and elsewhere: Don't italicize "spp".

Response: "spp" was de-italicized in line 103.

4. Line 105: Antibiotic resistant or phage resistant? Clarify.

Response: Specified "both antibiotic and phage resistant" in line 105.

5. Lines 117-124: You don't say if the 20 phages are novel or already known. If already published or their genome is in Genbank, you need to reference accordingly.

Response: These phages are novel and we have submitted their genomes to Genbank. Supplemental table S3 contains the links to their genomes. Line 123 now reads: "We isolated a collection of 18 additional *Enterococcus*-infecting phages from Southern California influent samples. Genome sequencing showed 8 *Myoviridae*, 10 *Siphoviridae*, and 1 *Podoviridae* phage. A list of the phages including one already known myovirus (EfV12-phi1) is shown in Supplemental Table S3, together with Genome size and Genbank accession numbers."

Table S3 figure legend now reads: The following phages were isolated and included in these experiments, with the phage name, Family, Genus, Genome size and Genbank accession numbers for the sequencing data we deposited, with the exception of EfV12-phi1, which was already available and which we used in our initial coevolution study Wandro et al 2019³¹. The naming convention for each phage begins with vB_OCPT for virus of bacteria and Orange County Phage Team (OCPT).

And the Data Availability section also refers to the genbank accession numbers in Table S3.

6. Line 125: Delete "Core". It must be "Genome alignments.." since you wouldn't know what is a core gene prior to aligning the genomes.

Response: "Core" was removed as suggested.

7. Line 140: Change "how phages" to "how select phages".

Response: Changed "how phages" to "how the phages we isolated".

8. Fig 2: Some numbers on the trees overlap the tree, making it hard to decipher. Move them off to one side. If necessary, cover them up and retype the number in a better location. Also, some font sizes are very tiny. Make larger.

Response: Thank you for your suggestions. We've modified figure 2 to improve readability.

9. Fig 4A: Are the lines in the graph showing the mean and SD of all of the phage treated cultures? For instance, is the line for the 1 phage cultures the mean and SD of all growth curves with 1 phage? This needs to be explained in the figure legend.

Response: Thank you for highlighting this ambiguity. Yes, we agree with your interpretation. The individual growth curves for each single phage are shown separately in Figure 4b in the OD600 panel, but in Fig 4a the 1 phage culture growth curves have averaged together. The caption for Figure 4a now reads: "Bacterial growth when *E. faecalis* Yi6-1 is grown alone (Host only), with each phage separately (1 Phage), or with phage cocktails (2 Phages, 3 Phages). The replicate mean is shown (OD₆₀₀ +/- SEM)."

10. Line 312: Are you saying the patient's names were de-identified? Presumably not, since that would be a HIPAA violation? Perhaps you mean the clinical isolates of *Enterococcus* were de-identified.

Response: By de-identified, we mean that all personal identifying information from the patients has been removed. We never had access to patient names as part of our study. Definition of de-identified from <https://www.edglossary.org/de-identified-data/> : “In education, **de-identified data** generally refers to data from which all **personally identifiable information** has been removed.” This is now around Line 312, which now reads: “*E. faecalis* and *E. faecium* strains were added to our strain collection from the patient population at UC San Diego Health; all personally identifying information was removed.”

11. Line 453: It appears you prepped and sequenced gDNA from the culture populations, instead of from individual colony-purified and confirmed phage-resistant strains. Is this true? If so, it should be stated that the cultures may contain multiple strains for clarification.

Response: Your observation is accurate; all sequencing experiments were performed on populations, not colony-purified strains. We have added the following statement starting on Line 468 of the manuscript to clarify this important point: “All sequencing experiments were performed using cultured populations of bacteria as opposed to individual colony-purified strains, therefore each culture likely contains DNA from multiple strain variations.”

May 9, 2022

Dr. Katrine Whiteson
University of California, Irvine
Department of Molecular Biology and Biochemistry
School of Biological Sciences
3236 McGaugh Hall
Irvine, CA 92697-3900

Re: mSystems00019-22R1 (Phage Cocktails constrain the growth of Enterococcus)

Dear Dr. Katrine Whiteson:

Your manuscript has been accepted, and I am forwarding it to the ASM Journals Department for publication. For your reference, ASM Journals' address is given below. Before it can be scheduled for publication, your manuscript will be checked by the mSystems production staff to make sure that all elements meet the technical requirements for publication. They will contact you if anything needs to be revised before copyediting and production can begin. Otherwise, you will be notified when your proofs are ready to be viewed.

Publication Fees:

We recognize that the video files can become quite large, and so to avoid quality loss ASM suggests sending the video file via <https://www.wetransfer.com/>. When you have a final version of the video and the still ready to share, please send it to mSystems staff at mSystems@asmusa.org.

For mSystems research articles, if you would like to submit an image for consideration as the Featured Image for an issue, please contact mSystems staff at mSystems@asmusa.org.

Sincerely,

Marta Gaglia
Editor, mSystems

Journals Department
Supplemental Table 3: Accept
Supplemental Table 2: Accept
Supplemental Figure S3: Accept
Supplemental Table S1: Accept
Supplemental Figure 2: Accept
Supplemental Figure 1: Accept
Supplemental Table S4: Accept